

# A past, present and future perspective on the European summer vapour pressure deficit

Daniel F. Balting[1*], Simon Michel[2], Viorica Nagavciuc[1,3], Gerhard Helle[4], Mandy Freund[5], Gerhard H. Schleser[6], David N. Steger[7], Gerrit Lohmann[1,8] and Monica Ionita[1,9]

[1] Alfred-Wegener-Institute, Bremerhaven, 27570, Germany

[2] Institute for Marine and Atmospheric research Utrecht (IMAU), Department of Physics, Utrecht University, Utrecht, Netherlands

[3] Faculty of Forestry, Ștefan cel Mare University, Suceava, 720229, Romania

[4] German Research Centre for Geosciences, Potsdam, 14473, Germany

[5] Climate and Energy College, University of Melbourne, Melbourne, VIC 3010, Australia

[6] Institute of Bio- and Geosciences IBG-3, Forschungszentrum Jülich, Jülich 52428, Germany

[7] Physics Department, University of Bremen, Bremen, 28359, Germany

[8] Deutsches Archäologisches Institut, Berlin, 14195, Germany

[9] Emil Racovita Institute of Speleology, Romanian Academy, Cluj-Napoca, 400006, Romania

Correspondence to: Viorica Nagavciuc (viorica.nagavciuc@awi.de)

**Abstract.** The response of evapotranspiration to anthropogenic warming is of critical importance to the water and carbon cycle. Conflicting observations about changes of evapotranspiration stem mostly from the brevity of observations in time and space as well as a high degree of internal variability. Here we present the first gridded reconstruction of the European summer vapour pressure deficit (VPD) for the past four centuries. The gridded reconstruction is based on 26 European tree-ring oxygen isotope records and is performed using a Random Forest approach. Based on our reconstruction, we show that from the mid-1700s a trend towards higher VPD occurred in Central Europe and the Mediterranean region which is based on the simultaneous increase in temperature and decrease in precipitation. This increasing VPD trend continues throughout the observational period and recent times. Climate model projections show this increase in VPD for the Mediterranean region continuing until the end of the 21st century, whereby the extent depends on the amount of greenhouse gas emissions. In contrast, projected VPD in North and Central Europe shows a tendency towards higher VPD only in the highest emission scenario. The reconstructed, observed and modelled VPD for past present and future is available here: https://doi.org/10.5281/zenodo.5958837 (Balting, D. F. et al., 2022).



## 1 Introduction

Evapotranspiration is a critical factor for understanding the links and feedbacks between atmospheric $CO_2$ and global climate (e.g., Good et al., 2015; IPCC, 2021). Within the terrestrial water fluxes, vegetation-produced transpiration represents the dominant factor (e.g., Jasechko et al., 2013; Good et al., 2015). One key driver for such vegetation resources and dynamics is vapor pressure deficit (VPD), which is defined by the difference between the water vapor pressure at saturation and the actual

water vapor pressure (Lawrence, 2005). VPD is a key variable for vegetation resources and dynamics (Grossiord et al., 2020), representing the atmospheric evaporative demand which has an influence on the leaf-level transpiration of plants and the corresponding stomatal conductance. With increasing VPD, stomata close to minimize water loss (Running, 1976) due to the high atmospheric evaporative demand. As consequence, a minimal stomata opening decreases stomatal conductance and photosynthetic activity (Fletcher et al., 2007). Extremely high VPD may even lead to reduced growth, a higher risk of carbon

starvation, and hydraulic failure (Grossiord et al., 2020). In contrast, low VPD leads to reduced water transport into the leaves and thus a reduced supply of nutrients. This marks VPD as an important indicator for plant activity (Novick et al., 2016), which among other things notably affects plant growth (Restaino et al., 2016), forest mortality (Park Williams et al., 2013), drought occurrence (Dai, 2013), crop production (Zhao et al., 2017) and wildfire occurrence (Seager et al., 2015).

Since VPD is a function of temperature (Lawrence, 2005), the effects of climate change and the associated rise in temperature

become evident for trends of VPD (Grossiord et al., 2020; IPCC, 2021). For instance, studies have shown that the water vapor pressure deficit has been increasing sharply at a global scale since the year 2000 (Simmons et al., 2010; Willett et al., 2014; Yuan et al., 2019). However, VPD records derived from remote sensing data cover only the last ~50 years and vary in quality, so that long-term perspectives of VPD are lacking. In addition, a long-term perspective can help to put recently observed trends of VPD in a long-term context and to estimate significance and robustness at local to continental scales. Furthermore, it is

essential to investigate the independent physiological effects of VPD on large-scale vegetation dynamics, which are less explored (Grossiord et al., 2020). So far, first local reconstruction studies have shown the potential of a long-term perspective on VPD (e.g., Liu et al., 2017). For example, Churakova-Sidorova et al. (2020) have shown that the recent VPD increase does not yet exceed the maximum values reconstructed during the Medieval Warm Anomaly in Siberia. Nevertheless, most studies lack a wider spatial perspective as they only reconstruct VPD time series for a location.

To obtain the first long-term and large-scale perspective on VPD dynamics, we use the stable oxygen isotope ratio of tree-ring cellulose ($\delta^{18}O_{cel}$). The use of $\delta^{18}O$ is motivated by the fact that the $\delta^{18}O_{cel}$ ratio is controlled by three main factors. The first factor is the $\delta^{18}O$ signature of precipitation ($\delta^{18}O_P$) supplying the trees with water through uptake by the roots from the soil. In middle and higher latitudes a significant positive relationship between $\delta^{18}O_P$ values and air temperature (commonly called the temperature effect) is observed (e.g., Rozanski et al., 2013). However, $\delta^{18}O_P$ variability in these latitudes cannot be explained

by air temperature alone (e.g. Welker (2000)). $\delta^{18}O_P$ at any location is also the result of the previous rainout history and origin(s) of the moist air mass(es). That is controlled by atmospheric circulation patterns (e.g., Edwards et al., 1996) and the sum of (temperature-dependent) oxygen fractionations occurring during evaporation and condensation of water that is finally



taken up by a tree. In the arboreal system, $\delta^{18}O$ of soil water ($\delta^{18}O_{SW}$) represents the $\delta^{18}O_{cel}$ input, that usually reflects an average $\delta^{18}O_P$ over several precipitation events. The average signal could be modified by fractionation due to partial evaporation of isotopically lighter water vapor from the soil (depending on soil texture and porosity; 04/02/2022 14:29:00). Therefore, $\delta^{18}O_{SW}$ in combination with $\delta^{18}O_P$ and $\delta^{18}O$ of the groundwater (depending on site and tree species) represents the baseline variability. The second factor is biochemical fractionation including partial isotopic exchange of cellulose precursors with stem water during cellulose biosynthesis (e.g. Saurer et al., 1997; Roden et al., 2000; Barbour, 2007), which is considered to be largely constant at 27±4 ‰ (Sternberg and Deniro, 1983).

The third factor, which is also the most important one, is the evaporative $^{18}O$ enrichment of leaf or needle water via transpiration of water vapor to the atmosphere (e.g., Saurer et al. (1997); Roden et al. (2000); Barbour (2007); Kahmen et al. (2011); Treydte et al. (2014) and citations therein). The transpiration process is controlled by leaf-to-air VPD modified by the aperture of stomata controlling the conductance for water vapor (Buckley, 2019). The $\delta^{18}O$ values of leaf water are typically enriched in $^{18}O$ compared to the plant's parent water because evaporative losses are greater for the lighter $^{16}O$ than for $^{18}O$ (Roden et al., 2000). However, $\delta^{18}O_{cel}$ features the $\delta^{18}O$ signature of chloroplast water which is not in isotopic equilibrium with leaf water at the actual sites of transpiration (stomata), i.e. the higher the transpiration rates the lower is the rate of enrichment of the chloroplast water (Péclet effect, Barbour et al. (2004)). Nonetheless, since the variability of $\delta^{18}O_{cel}$ results predominantly from a combination of the temperature-dependent $\delta^{18}O$ of the water source and the evaporative $^{18}O$ enrichment of leaf water controlled by leaf-to-air VPD, $\delta^{18}O_{cel}$ can be used as a proxy for variations in VPD (Ferrio and Voltas, 2005; Kahmen et al., 2011).

To gain a spatial and temporal perspective on past VPD variability, we use multiple stable oxygen isotope records derived from tree-rings and apply a Random Forest (RF, Breiman 2001) regression method, which is often considered as one of the core approaches for machine learning. According to Yang et al. (2020), RF has become one of the most successful machine learning algorithms for practical applications over the last two decades due to its proven accuracy, stability, speed of processing, and ease of use (Rodriguez-Galiano et al., 2012; Belgiu and Drăguţ, 2016; Maxwell et al., 2018; Bair et al., 2018; Qu et al., 2019; Reichstein et al., 2019; Tyralis et al., 2019). The term RF describes a non-linear and robust technique in which several decision trees are built and aggregated at the end to make predictions or perform reconstructions (Breiman, 2001). The RF approach is increasingly applied in climate and environmental sciences, and has been used for the prediction of snow depth (Yang et al., 2020), solar radiation (Prasad et al., 2019), daily ozone (Zhan et al., 2018), precipitation (Ali et al., 2020); as well as reconstructions of last millennium North Atlantic Oscillation (Michel et al., 2020), streamflow since 1485 C.E. (Li et al., 2019) and vegetation cover during the mid-Holocene and the Last Glacial Maximum (Lindgren et al., 2021). Although RF models have proofed to be a useful method in geosciences, studies on the spatial-temporal reconstruction of climate variables based on $\delta^{18}O_{cel}$ are relatively rare due to the low availability of time series.

In this study, we use a European $\delta^{18}O_{cel}$ network from tree rings, consisting of 26 sites that cover the period 1600 – 1994 (Balting et al., 2021) and a RF approach (Breiman, 2001) to reconstruct gridded fields of European summer VPD from 1600 to 1994. Our goal is to provide a spatial and long-term perspective on the past VPD variability. Therefore, the reconstruction is analysed from a spatio-temporal perspective for the regions Northern Europe, Central Europe, and Mediterranean (Iturbide





et al., 2020) as defined and used in the Sixth Assessment Report of the IPCC (IPCC, 2021). In this respect, the paper is structured as follows: in Section 2 we give a detailed description of the data and methods employed throughout the paper, while the main results are presented in Section 3. The spatial variability of VPD is investigated for selected extreme years, for

example, the summer after the Tambora eruption (*e.g.*, 1815-1816 A.D.). Afterwards, the reconstruction is compared to other studies, and uncertainties are discussed. Furthermore, we compare the VPD observations with climate model projections of low (Shared Socioeconomic Pathways 1-2.6; SSP1-2.6), medium (SSP2-4.5), and high (SSP5-8.5) emission scenarios. In this way, the past and present VPD conditions are evaluated to assess the statistical significance of observed trends and which can be used to understand VPD variability on a local, regional and continental scale. The main conclusion is shown in Section 4.

## 105 2 Sample sites and used climate data

### 2.1 The stable isotope network

To reconstruct the European summer VPD, we use 26 time series of stable oxygen isotope in tree-ring cellulose (Figure 1). 21 time series of the 26 were obtained from the dataset generated by the EU project ISONET (EVK2-CT-2002-00147) (e.g. Treydte et al., 2007b, a; Balting et al., 2021). In addition to the ISONET dataset, we added five new time series of stable ox-

ygen isotopes in tree-ring cellulose from Bulgaria, Turkey, southwestern Germany, Romania, and Slovenia (Heinrich et al., 2013; Hafner et al., 2014; Nagavciuc et al., 2018).

The detailed measurement methodologies used within the ISONET project as well as for the other sites are described in Boettger et al. (2007), Treydte et al. (2007a, b), Heinrich et al. (2013), Hafner et al. (2014), and Nagavciuc et al. (2018), respectively. At least four dominant trees were selected at each site and two increment cores were taken per tree for the ISONET

project; 15 cores of five living trees were taken from the site in Turkey (Heinrich et al., 2013), 12 trees were sampled for the Slovenian time series (Hafner et al., 2014) and from nine trees, one core per tree was taken in Romania (Nagavciuc et al., 2018). Standard dendrochronoluac dating method was performed (Fritts, 1976), and subsequent individual growth rings were dissected from the cores. All tree rings from the same year were pooled for most sites prior to cellulose extraction for the ISONET sites (Treydte et. al., 2007a, b) as well as for the Romanian site (Nagavciuc et al., 2018). The dissected tree rings

from the Slovenian and Turkish sites were measured individually and not pooled (Heinrich et al., 2013; Hafner et al., 2014). For oak, only the latewood was used for the analyses, because this approach was assumed to use predominantly climate signals from the current year, as the earlywood of oaks usually contains climate information from the previous year (González-González et al., 2015; Davies and Loader, 2020). The results are expressed using the conventional δ (delta) notation, in per mil (‰) relative to the Vienna Standard Mean Ocean Water (VSMOW; Craig, 1957).

Our isotope network consists of nine deciduous tree sites (*Quercus*) and 17 conifer sites *(Pinus, Juniper, Larix*; see Table 1). The sample sites are well distributed over Europe (Fig. 1). The elevation of the locations varies from 10 m a.s.l. (Woburn) to 2.120 m a.s.l. (Pedraforca). The longest chronologies cover a period from 1600 to 2005. The highest data density is available

for the period 1900-1994 with 26 time series (Fig. 2). For several sites or regional groups of sites from the ISONET datasets,
the data is published within individual studies (Saurer et al., 2008; Vitas, 2008; Etien et al., 2008; Hilasvuori et al., 2009; Haupt

et al., 2011; Saurer et al., 2012; Rinne et al., 2013; Helama et al., 2014; Labuhn et al., 2014; Saurer et al., 2014; Labuhn et al.,
2016; Andreu-Hayles et al., 2017).

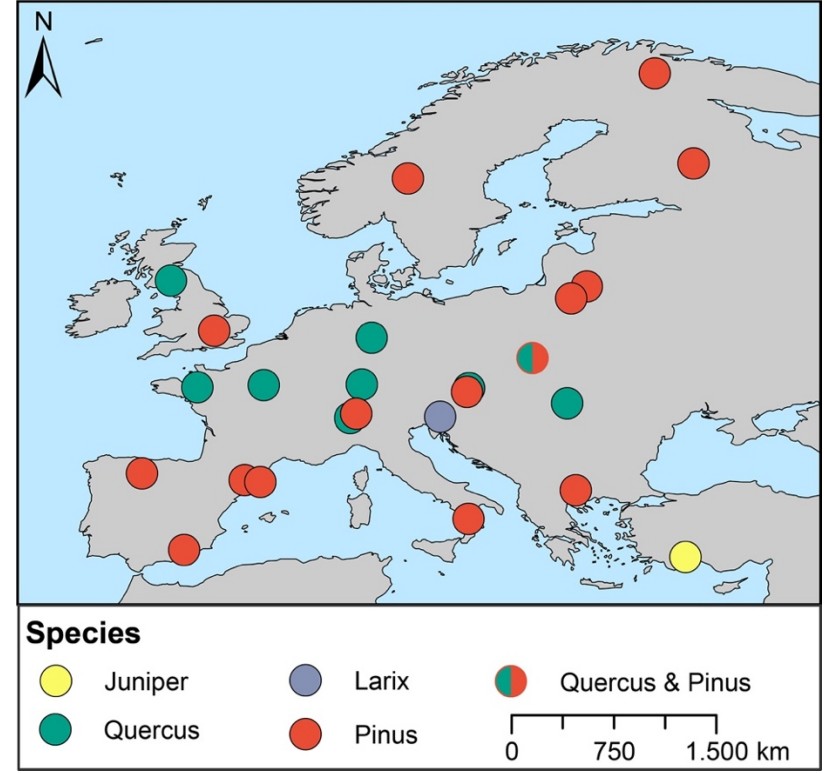

**Figure 1: The site distribution of the used $\delta^{18}O_{cel}$ network combined with the corresponding tree species.**

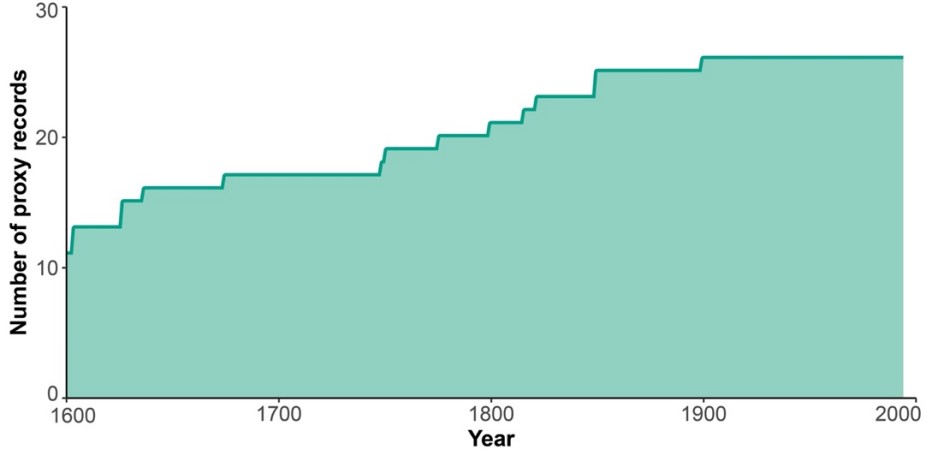

**Figure 2: The number of available $\delta^{18}O_{cel}$ time series within the network.**





**Table 1: Characteristics of each sample site used within our study.** 21 of the 26 $\delta^{18}O_{cel}$ records were obtained from the EU project ISONET (Treydte et al. 2007a, Balting et al. 2021) and five additional sites from Bulgaria, Turkey, Southwest Germany, Romania, and Slovenia were added (Hafner et al., 2014; Heinrich et al., 2013; Nagavciuc et al., 2018).

| Location | Country | Species | First year | Last year | Lon. | Lat. | Altitude |
|---|---|---|---|---|---|---|---|
| Cazorla | Spain | Pinus nigra | 1600 | 2002 | -2.57° | 37.53° | 1820 m |
| Cavergno | Switzerland | Quercus petraea | 1637 | 2002 | 8.36° | 46.21° | 900 m |
| Dransfeld | Germany | Quercus petraea | 1776 | 2002 | 9.78° | 51.50° | 320 m |
| Fontainebleau | France | Quercus petraea | 1600 | 2000 | 2.67° | 48.38° | 100 m |
| Gutuli | Norway | Pinus sylvestris | 1600 | 2003 | 12.18° | 62.00° | 800 m |
| Inari | Finland | Pinus sylvestris | 1600 | 2002 | 28.42° | 68.93° | 150 m |
| Isibeli | Turkey | Juniper excelsa | 1850 | 2005 | 30.45° | 37.06° | 1800 m |
| Lainzer Tiergarten | Austria | Quercus petraea | 1600 | 2003 | 16.20° | 48.18° | 300 m |
| Lochwood | United Kingdom | Quercus petraea | 1749 | 2003 | -3.43° | 55.27° | 175 m |
| Monte Pollino | Italy | Pinus leucodermis | 1604 | 2003 | 16.16° | 39.58° | 1900 m |
| Mount Vichren | Bulgaria | Pinus heldreichii | 1800 | 2005 | 23.24° | 41.46° | 1900 m |
| Naklo | Slovenia | Larix decidua | 1600 | 2005 | 14.30° | 46.30° | 440 m |
| Niepolomice | Poland | Quercus robur & Pinus sylvestris | 1627 | 2003 | 20.38° | 50.12° | 190 m |
| Nusfalau | Romania | Quercus robur | 1900 | 2016 | 22.66° | 47.19° | 270 m |
| Panemunes | Lithuania | Pinus sylvestris | 1816 | 2002 | 23.97° | 54.88° | 45 m |
| Pedraforca | Spain | Pinus uncinata | 1600 | 2003 | 1.42° | 42.13° | 2120 m |
| Pinar de Lillo | Spain | Pinus sylvestris | 1600 | 2002 | -5.34° | 42.57° | 1600 m |
| Plieningen | Germany | Quercus petraea | 1822 | 1999 | 9.13° | 48.42° | 340 m |
| Poellau | Austria | Pinus nigra | 1600 | 2002 | 16.06° | 47.95° | 500 m |
| Rennes | France | Quercus robur | 1751 | 1998 | -1.7° | 48.25° | 100 m |
| Sivakkovaara | Finland | Pinus sylvestris | 1600 | 2002 | 30.98° | 62.98° | 200 m |
| Suwalki | Poland | Pinus sylvestris | 1600 | 2004 | 22.93° | 54.10° | 160 m |
| Vigera | Switzerland | Pinus sylvestris | 1675 | 2003 | 8.77° | 46.50° | 1400 m |
| Vinuesa | Spain | Pinus uncinata | 1850 | 2002 | 2.45° | 42.00° | 720 m |
| Woburn | United Kingdom | Pinus sylvestris | 1604 | 2003 | -0.59° | 51.98° | 10 m |



**2.2 Observational data**

Mean surface temperature in °C and near surface relative humidity in Vol. % are derived from the 20th Century Reanalysis Project (20CR) version V3 (Slivinski et al., 2019) at a monthly resolution. The 20CR reanalysis has a temporal resolution of three hours, 28 different pressure levels, and a resolution of 1° x 1°. We use the ensemble mean derived from an 80-member ensemble. The climate variables are available for the period from 1836 to 2015 and they are provided by NOAA/OAR/ESRL PSL, Boulder, Colorado, USA (https://psl.noaa.gov/data/gridded/data.20thC_ReanV3.html).

**2.3 General procedure of the European VPD reconstruction**

We have used the Random Forest method (RF; see Supplement Section 1; Breiman, 2001) to reconstruct European summer VPD spatially for the last 400 years. The VPD data derived from 20CRV3 (Slivinski et al., 2019) and the $\delta^{18}O_{cel}$ network data are used as input. There is a small number of missing data (0.38% entries in total) which are infilled following Josse and Husson (2016). Before the reconstruction is started, a sensitivity study is done with the $\delta^{18}O_{cel}$ network to ensure the relation

between $\delta^{18}O_{cel}$ and observed VPD with correlations ($\alpha < 0.05$).

The reconstruction is based on the methodology presented by Michel et al. (2020) and the corresponding scripts (ClimIndRec version 1.0), which were reprogrammed for a spatial reconstruction. In our study, we focus on the continental area of the European region (34.5°W to 49.5°E and 30.5°N to 74.5°N). The testing and validation period is set from 1900 to 1994 (Fig. 2). We use a nesting approach for our study which has the advantage that time series with different temporal coverage can be

integrated into the reconstruction. With additional time series being available multiple reconstructions are derived. At the end of the calculations, the data sets are aggregated so that the RF models with the highest data coverage represents the final VPD reconstruction. In addition to the reconstruction, validation parameters are calculated for each run of the RF reconstruction using the Coefficient of Efficiency metric (CE; Supplement Section 1; Nash and Sutcliffe, 1970).

Finally, we compare the regional averages and running averages (30 years) of the reconstructed VPD with temperature

(Luterbacher et al., 2004), precipitation (Pauling et al., 2006), and Palmer Drought Severity Index (PDSI) reconstructions (Cook et al., 2015) for the three European AR6 regions: Northern Europe, Central Europe and Mediterranean (Iturbide et al., 2020). Furthermore, we show maps of the European summer VPD for selected and highlighted wet (1737, 1814, 1815) and dry (1616, 1741, 1821) years to investigate the spatial variability of our reconstruction (Brooks and Glasspoole, 1922; Pauling et al., 2006; Trigo et al., 2009; Brázdil et al., 2013; Cook et al., 2015; Ionita et al., 2021). Before this mapping, all VPD grid

cells are centred and standardized (z-transformation) to present z-anomalies for each grid cell.

**2.4 VPD computation and further pre-processing**

The calculation of the monthly VPD for Reanalysis, historical, and future climate projections are based on the study of Barkhordarian et al. (2019) where the near surface air temperature (T) and the dew point temperature ($T_d$) both in °C are used.



Since dew point temperature is not available for each of the datasets, we have used T and relative humidity (RH) to compute
$T_d$ as follows:

$$T_d = \frac{a_1 * (\ln\left(\frac{RH}{100}\right) + \frac{a_2 * T}{a_1 + T})}{a_2 - (\ln\left(\frac{RH}{100}\right) + \frac{a_2 * T}{a_1 + T})}$$

Where $a_1$ and $a_2$ are defined as $a_1 = 243.04$ °C and $a_2 = 17.625$ hPa. This computation is reliable and is notably used in many
climate models (Barkhordarian et al., 2019). We utilize the Clausius–Clapeyron relation applying a term for the saturation
vapor content of the air and a term for the actual vapor pressure to calculate VPD as follows (Marengo et al., 2008; Seager et
al., 2015; Barkhordarian et al., 2019; Behrangi et al., 2016):

$$VPD = c_1 \times e^{(\frac{c_2 * T}{c_3 + T})} - c_1 \times e^{(\frac{c_2 * T_d}{c_3 + T_d})}$$


where c1 = 0.611 kPa, c2 = 17.5, c3 = 240.978 °C and VPD is in kPa (see WMO (2018) for further information). We compute
seasonal averages DJF (December to February, winter), MAM (March to May, spring), JJA (June to August, summer), and
SON (September to November, fall).

**2.5 Future scenarios**

Monthly mean surface temperature and near surface relative humidity from the latest CMIP 6 models (Eyring et al., 2016) are
used to compute VPD until the end of the 21[st] century. We use the historical simulations (1850 to 2014) in conjunction with
the future simulations based on the shared socioeconomic pathways (SSPs) projections (O'Neill et al., 2016) for the years 2015
to 2100. We decided to focus on the SSP1-2.6, SSP2-4.5, and SSP5-8.5 scenarios. The criteria for the model selection is as
follows (Seneviratne and Hauser, 2020): i) the selected models must provide the relevant variables, (ii) the models must have
a temporal coverage from 1850 to 2100, and (iii) the selected models do not have duplicate or missing time steps. The presented
regional means are calculated from the original grid resolution; for the maps, the data is regridded bilinear to a 1°x 1°grid. All
models are listed in Table S1.

We compare the regional averages of the reconstructed and observed VPD with the projected VPD for three European AR6
regions (Iturbide et al., 2020). Beside the temporal perspective, we will show anomaly maps to investigate spatial trends in
future scenarios.



# 3 Results

## 3.1 Climate sensitivity

The network of $\delta^{18}O_{cel}$ time series is significantly correlated with the European Summer VPD over the observational period (i.e., 1900 – 1994). In Figure 3, we show the correlations of the individual $\delta^{18}O_{cel}$ time series with the respective time series of the grid cell from the calculated VPD of 20CRV3 (Slivinski et al., 2019) for summer and spring.

In total, 11 sites show a significant correlation with VPD in spring, with the maximum in Italy and Turkey (r=0.40, Fig. 3a). The sites with significant correlations are distributed across the continent without any spatial bias. There is no influence of the tree species used and no influence of the altitude of the sampled trees (Balting et al., 2021). The high sensitivity of the time series from Turkey with spring climate has already been noted (Heinrich et al., 2013). 15 time series out of the 26 sites show no significant correlations with VPD during spring.

The most significant correlations as well as the highest seasonal correlation coefficients are reached with the summer VPD (Fig. 3b). 22 of the 26 sites show a significant correlation, with most of these sites being located in France, Spain, Germany, Scandinavia, and Great Britain. The highest correlation coefficient (r=0.64) was obtained for the Rennes site (France). One time series from Turkey, two from Spain, and one from Austria do not have a significant correlation with the summer VPD (Fig. 3b). Based on these results and the shown sensitivity of the $\delta^{18}O_{cel}$ for VPD variability, we perform the reconstruction of VPD for the summer months (JJA).

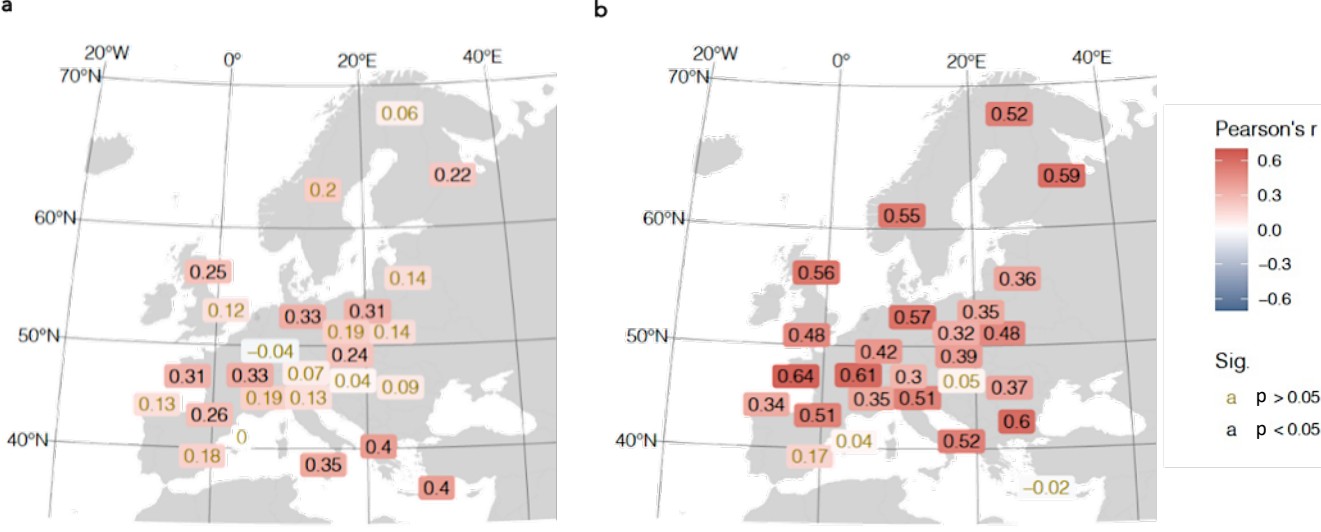

**Figure 3: Correlation of the $\delta^{18}O_{cel}$ time series with the European VPD for the period 1900 to 1994. a,** for spring (MAM), **b,** for summer (JJA). The significant correlations are highlighted (p<0.05).





### 3.2 Validation statistics

Since we used the nesting approach in our study, we calculated and optimised by cross validation (see Michel et al. (2020)) 13 RF models in total, i.e., from each time when a new time series from the network is available. The quality of the reconstruction can be described with the CE (Supplement Section 1), which is presented in Figure 4 for four selected time steps. For the year 1600, CE scores indicate that a satisfactory quality for the reconstruction of VPD for Northeast Spain, Italy, Greece, France, Germany, and large parts of Scandinavia (Fig. 4). The described regions coincide with the locations of the eleven available time series for this time slice. Therefore, our reconstruction covers large parts of Europe from the beginning. For the time slice 1700 presented in Figure 4, an expansion of the significant regions is observed for the CE values. For this time slice, 16 of the 26 time series are available and the spatial coverage has improved towards Eastern Europe, and isolated gaps, such as in parts of Norway, could be closed. A further improvement of the CE values between 1700 and 1800 can also be observed (Fig. 4), whereby it is most noticeable that Great Britain is now largely covered by grid points robustly reconstructed. This is because another time series from Scotland has been included for the 1700 time slice (Fig. 4). Also, for the last time step an improvement of the spatial coverage of the reconstruction can be seen. In particular, a large gap in Eastern Europe has been closed, due to the time series from Romania, which is available for the reconstruction from 1900 onwards. It can also be noted that adjacent regions of Europe, such as parts of Turkey also have a suitable quality for reconstructing past VPD variability changes. The overall spatial strength of the network is in Southern, Western, and Northern Europe, whereas Eastern Europe can only be partially covered. Nevertheless, it must be mentioned that validation statistics for all time steps must be included in the consideration of the results. Therefore, only those grid cells with a satisfactory reconstruction performance will be included in the following analyses.

### 3.3 Temporal variability of the European VPD reconstruction

The average VPD of Northern Europe (Fig. 5a) shows significant differences in variability as well as in the long-term mean. Northern Europe shows the lowest VPD of all three regions, which is due to the comparably low temperatures and high humidity in these broad areas. The increase of VPD between 1620 and 1660 is significant (m = 0.0017 kPa.year$^{-1}$; p-value= 0.0026, where m is the linear regression slope), with the highest VPD value of the 400 years being reached in 1652 (VPD= 0.4619 kPa > 2.7 × σ, where σ is the standard deviation of the time series). After this peak, VPD drops until it starts to rise again in 1673. However, we note that this increase during the Late Maunder Minimum (1675-1715; considered as the coldest phase of the Little Ice Age in Europe (Brönnimann, 2015)) is not as long-lasting as the former increase from 1620 to 1660. VPD starts to decrease again from the year 1700. The subsequent drop lasts until the 1720 to 1730 period, after which the time series shows further low-frequency variability and a minor upward tendency in the 30-year rolling average that lasts until the end of the 20th century (m= 0.00002 kPa.year$^{-1}$; p-value < 0.00001). In total, the three years with the lowest VPD in this region are 1674, 1782, and 1802 whereas the years with the highest VPD are 1652, 1735, and 1959.



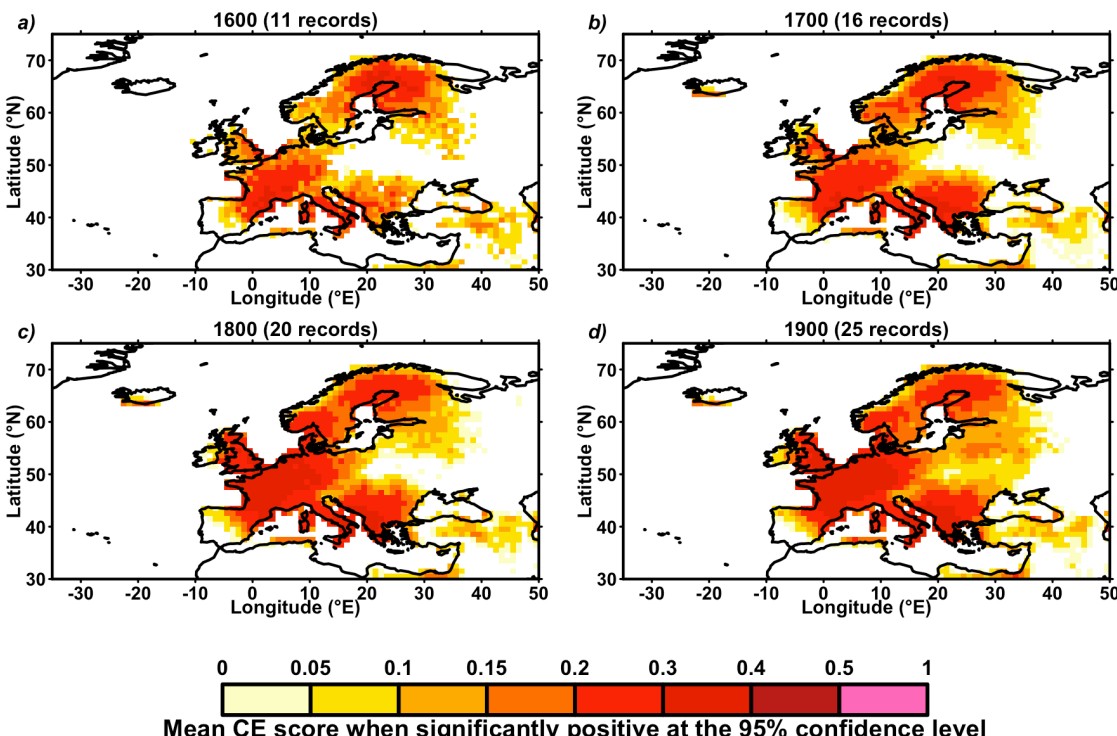

**Figure 4: The Nash–Sutcliffe CE for four different time slices.** The coloured areas show the regions where the model set up has a suitable quality, which was tested with a one-sided Student's t-test (p<0.05). No or lack of model efficiency largely relates to regions without $\delta^{18}O_{cel}$ records (e.g. SW Spain, Belarus, Ukraine).

The VPD reconstruction for Central Europe (Fig. 5b) shows that VPD increases from 1600 up to the Late Maunder Minimum (*i.e.* early 1700s) (m= 0.0007 kPa.year$^{-1}$; p-value = 0.0003). The maximum of the 30-year rolling time series is reached in 1697 (VPD = 0.6398 kPa > 2.4 × σ). This increase is followed by a downward trend of VPD in Central Europe which ends in 1743 (m= -0.0014 kPa. year$^{-1}$; p-value= 0.0286). The period 1740 to 1760 is characterized by very low VPD values and the lowest 30-year rolling VPD. From this time on, the rolling average VPD is characterized by a rising trend (m= 0.0002 kPa.year$^{-1}$; p-value= 0.0037), but low values are reached during the Dalton Minimum (1790 to 1820). Furthermore, the VPD reconstruction is characterized by a significant 60 to 80 years oscillation, where the lowest values are reached during the periods 1890 to 1920 and 1960 to 1980 (Supplement). In total, the three years with the lowest VPD in this region are 1602, 1755, and 1786 whereas the years with the highest VPD are 1707, 1835, and 1921.

The VPD reconstruction for the Mediterranean region shows the largest VPD and higher variability compared to the other two regions (Fig. 5c). It is noticeable that at the beginning of the reconstruction period (1610-1650) the VPD values decrease (m= -0.0017 kPa. year$^{-1}$; p-value= 0.013), which is followed by a VPD increase leading to an almost constant VPD level during

the Late Maunder Minimum. From the end of the Maunder Minimum, the VPD falls steadily until 1761, where it remains at a similar level on average until 1773. This is followed by a short increase, which is stopped at the beginning of the Dalton

Minimum and leads to a decreasing VPD. The minimum of VPD value is also reached during this period (VPD=0.96 kPa for the year 1814). After the Dalton Minimum, the VPD rises again (m= 0.0003 kPa.year$^{-1}$; p-value= 0.0182). Furthermore, the VPD time series shows similar oscillations as for Central Europe, but most pronounced in the last 150 years. Low periods of the oscillation are shown for example between the end of the 19th century and the beginning of the 20th century as well as for the 1960s and 70s. In total, the three years with the lowest VPD in this region are 1780, 1814, and 1815 whereas the years with

the highest VPD are 1686, 1874, and 1945.

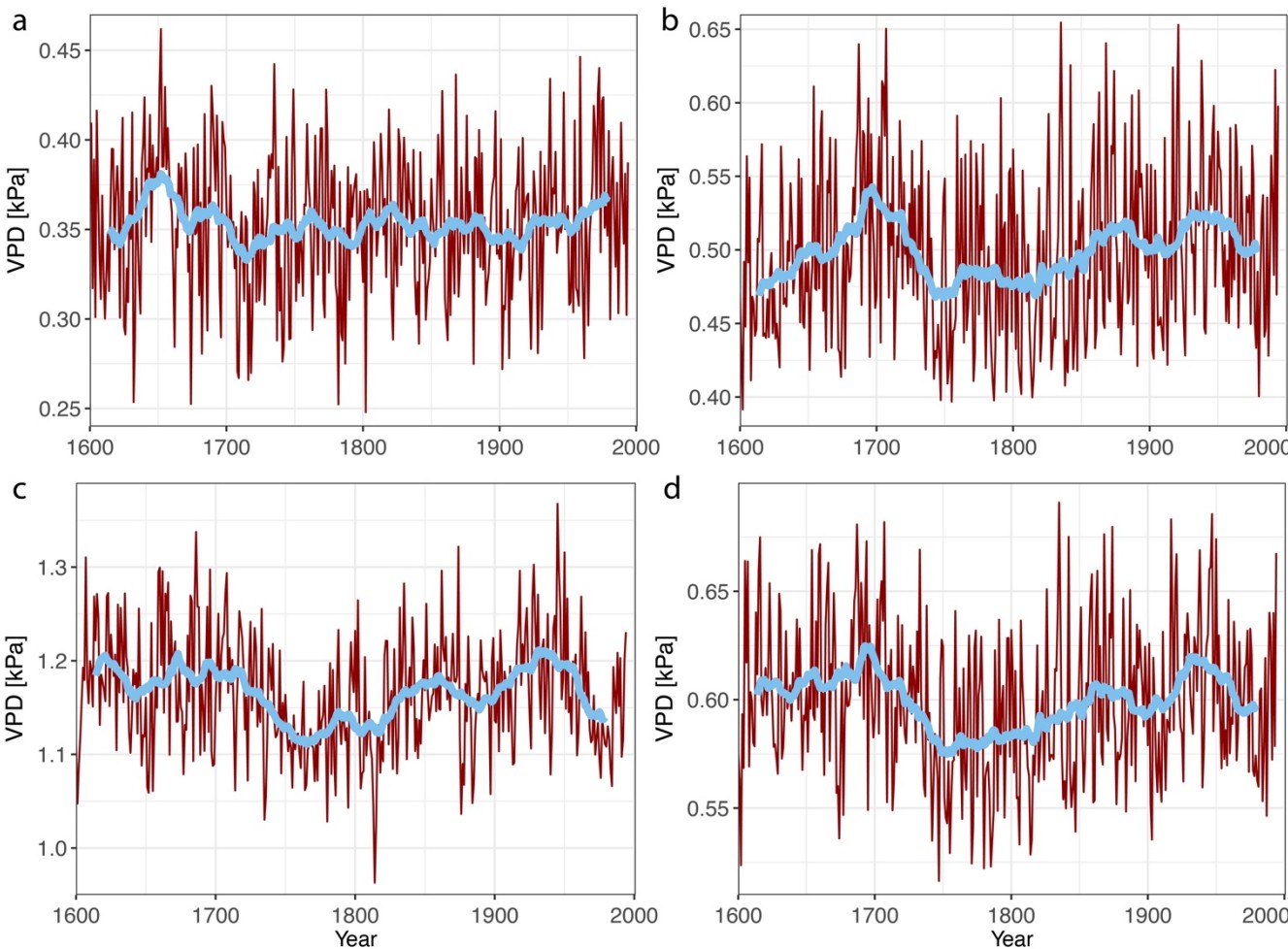

**Figure 5: Temporal variability of the VPD reconstruction. a,** Northern Europe, **b,** Central Europe, **c,** Mediterranean region, **d,** Europe. The red curve represents the annual summer values and the blue line the rolling mean for 30 years window. The average calculation included only those grid cells that had a sufficient CE in the validation statistics.





The mean VPD value for the entire Europe increases until about 1690 (Fig. 5d). However, the maximum VPD for Europe is reached before the Late Maunder Minimum. During the Late Maunder Minimum, the trend starts to reverse and the average VPD decreases until the middle of the 18th century. From there on, the European mean VPD increases (m= 0.0001 kPa. year$^{-1}$; p-value= 0.00006), again due to the positive trends in Central Europe as well as changes in the Mediterranean.

**3.3 Comparison of the temporal perspective with other reconstructions**

We compare the VPD variability with existing reconstructions of summer temperature (Luterbacher et al., 2004), precipitation (Pauling et al., 2006), and drought (PDSI) reconstruction (Cook et al., 2015) in Figure 6, 7, and 8.

For Northern Europe, the characteristics of VPD until 1700 show little resemblance with the reconstructions of precipitation and temperature (Fig. 6a,b,c). However, the PDSI indicates dry conditions for the time of the maximum VPD in 1652 (Fig. 6d). Furthermore, the decrease of VPD between 1700 to 1730 is shown as a wet period in the drought reconstruction, which

could be a possible explanation for these low values. From 1730 to 1800, the shown strong temperature increase is visible in VPD only partly. A possible reason for this could be the concurrent increase in precipitation. Since the temperature and precipitation changes in this region coincide, it is difficult to detect the characteristics of the two variables in the VPD time series. For Central Europe, the increase of the VPD until 1700 cannot be explained by the temperature or precipitation reconstruction (Fig. 7a,b,c). The 30-year running means of both time series show diverging trends. Nevertheless, the reconstruction of the

PDSI also shows a trend towards drier conditions from 1670 to 1690 (Fig. 7d). However, before the end of the 16th century, the data indicates a wetting trend again. Afterwards, the periods of high and low temperature and precipitation match very well with the corresponding periods of high and low VPD. For example, the decrease in precipitation and temperature between 1730 and 1745 is also present in the VPD time series. From 1800 onward, the VPD increases with a clear and significant oscillational behaviour which is closely related to the oscillational pattern of temperature (Supplement Figure S2). However,

the oscillation of the VPD time series is also influenced by the precipitation variability. The precipitation shows an oscillating behaviour from the second half of the 19th century, but the magnitude is represented weaker than in the VPD or the temperature time series. While the temperature, precipitation, and VPD reconstruction show a consistent picture from 1800 onwards, meaning that it becomes warmer, there is less precipitation, and the VPD increases in summer. The PDSI, on the other hand, shows a wetting trend from 1800 onwards. We suggest that this could be based on accumulation processes of precipitation in the

preceding months which is represented by the PDSI. This could compensate for the increasing temperature and the decreasing precipitation in summer.

For the Mediterranean area, the summer of 1814 is an extreme event in the temperature time series, as it shows the second coldest temperatures for this region in the last 400 years (Fig. 8a,b,c). Furthermore, it is interesting that the precipitation increased during the Dalton Minimum (~1790) in this region and only decreases towards its end (1805-1810), which can also

explain the behaviour of the VPD during this period. We suggest that the comparable low temperatures and the high precipitation led to this extreme year in the VPD time series. After the Dalton Minimum, the temperature shows an increasing trend

Earth System
Science
Data

and the course of the curves is very similar to the oscillating behaviour already described for the VPD, even though the magnitude differs. Nevertheless, the periods with high and low values of temperature are almost identical to the corresponding periods of the VPD time series after the Dalton Minimum. Since the decrease in precipitation is almost linear in the 30-year

mean from 1850, we suggest that, comparatively, the oscillation of temperatures can be clearly observed in the VPD time series.

**Figure 6: The reconstructed summer VPD for Northern Europe in comparison with reconstructed summer temperature, precipitation, and PDSI. a,** reconstructed VPD, **b,** surface temperature (Luterbacher et al., 2004), **c,** precipitation (Pauling et al., 2006), **d,** PDSI

(Cook et al., 2015). The boundaries of the region are following the definitions of Iturbide et al. (2020). The red line represents the annual summer values and the blue line the rolling mean for 30 years window. The average calculation included only those grid cells that had sufficient CE in the validation statistics.



A similar situation is also described by the drought reconstruction for the Mediterranean area (Fig. 8d). Thus, the reconstructed PDSI shows wetter conditions towards the middle of the 18th century, which is complement by the low VPD values for this

period. Like the VPD and the temperature reconstruction, the PDSI shows drier conditions after the 1770s, intensifying until the beginning of the Dalton Minimum (~1790). Furthermore, the year 1814 is also the year with the wettest conditions in the PDSI time series. After the Dalton Minimum, the anti-correlation between temperature/VPD and the PDSI is again evident, so that an oscillatory behaviour is also visible.

**Figure 7: Same as Fig. 6, but for Central Europe.**

**Figure 8: Same as Fig. 6, but for the Mediterranean region.**

## 3.4 Spatial variability of the VPD reconstruction

To illustrate the spatial variability of the VPD reconstruction, we selected extreme years that represent either a particularly wet

or dry period (see Fig. 9). The first presented pattern of the spatial VPD variability is for the summer of the year 1616 (Fig. 9 upper left). For this year, our VPD reconstruction represents positive VPD anomalies over the entire continent pattern where the high anomalies are reached in Germany, Austria, parts of Czech, and Switzerland. The highest VPD anomaly is shown in





Scotland (VPD$_{z-anomaly}$ = -3.1). Therefore, the spatial pattern of the VPD anomalies for the year 1616 shows similarities to a monopole pattern over Europe, since no negative VPD anomalies are shown.

The VPD reconstruction for the summer of 1737 shows negative VPD anomalies for Central and Southern Europe (Fig. 9 upper-mid), where the lowest anomalies are shown for the Balkan area (VPD$_{z-anomaly}$ = -2.16). In contrast, Finland, northern Sweden, and northern Norway are characterized by high positive VPD anomalies (maximum VPD$_{z-anomaly}$ = 2.92). The pattern can be described as a dipole pattern between northern Scandinavia and Central/South Europe.

The VPD anomaly for the summer of 1741 also presents a dipole pattern (Fig. 9 upper right), but the centres are in this case

Scandinavia and Southwest Europe. Low VPD anomalies are located in Scandinavia with the lowest values in southern Sweden and Denmark (VPD$_{z-anomaly}$ = -2.73). In contrast, Southwest Europe is characterized by high VPD anomalies with the highest values located in northeast Spain (VPD$_{z-anomaly}$ = 2.47). Also, Southeast Europe shows positive VPD anomalies, but with a lower magnitude compared to Northeast Spain.

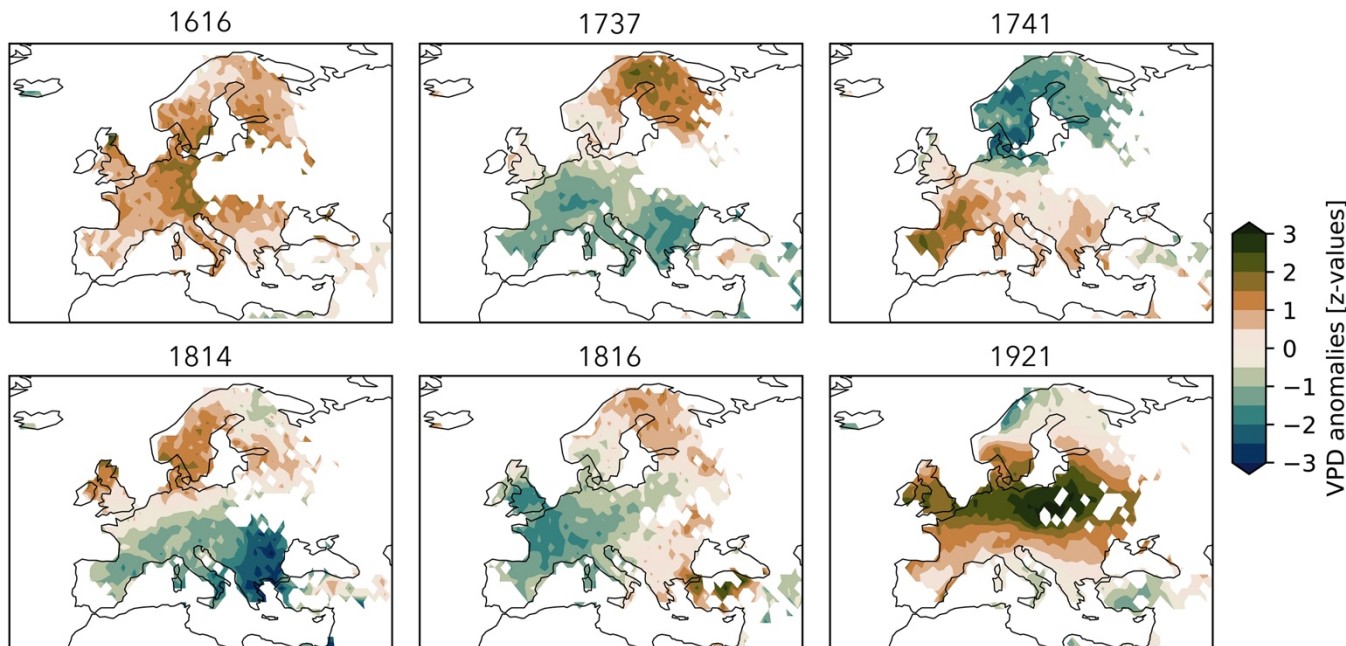

**Figure 9: Spatial pattern of VPD anomalies for selected years.** Values of each grid cell are standardised and centred. The anomalies were computed based on the subtraction of the average of the twelve preceding years as well as the twelve following years. Furthermore, only those grid cells with sufficient CE in the validation statistics are included in the plot.

The year 1814 is the year with the lowest VPD average in the Mediterranean region (Fig. 5). The VPD anomaly map (Fig. 9 lower left) shows a dipole between Northern and Southern Europe for this year. The highest VPD anomalies are shown in

West Sweden, West Norway, Denmark, and Great Britain with the highest values in southern Sweden and Denmark (VPD$_{z-anomaly}$ = 2.09). The opposite situation is presented for the entire Southern Europe, where negative VPD anomalies characterize the entire area. The lowest VPD values are located in Southeast Europe (VPD$_{z-anomaly}$ = -3.51).





The VPD anomaly map for 1816 illustrates a dipole between West/East (Fig. 9 lower mid). Negative VPD anomalies are shown for West and Central Europe with the lowest values in Great Britain and France ($VPD_{z-anomaly}$ = -2.06). In contrast,

positive values are represented in East Europe with the highest values in Scandinavia and Southeast Europe especially Greece and Turkey ($VPD_{z-anomaly}$ = 3.16).

A tripolar pattern is shown in the VPD anomaly map for the year 1921 (Fig. 9 lower right). High VPD values are observed for Central Europe which stretch in a band from North Spain/Great Britain to Poland. The highest VPD anomalies are located in eastern Germany, Poland, and Lithuania ($VPD_{z-anomaly}$ = 3.39). In contrast, negative VPD anomalies are shown in southern

Italy, Turkey, and northern parts of Scandinavia. The lowest values are presented in Norway ($VPD_{z-anomaly}$ = -2.15).

These spatial variations and varying VPD patterns underline the importance of a spatial field reconstruction.

### 3.5 Comparison of past/historical spatial variability

According to a drought reconstruction for Czech, the drought of 1616 was one of five "outstanding drought events" since 1090 CE which began in the spring and continued throughout the summer with great heat and dried-up rivers (Brázdil et al., 2013).

The dry conditions are also represented by our reconstruction, where positive VPD anomalies are shown for Europe with the highest values are reached in Germany, parts of Czech, Austria, and Switzerland (Fig. 9 upper left). In addition, the average time series for Europe in Figure 5 also indicates high VPD values during this time. A similar situation is also described by the PDSI (Cook et al., 2015), where severe to extreme drought are shown over Central and Eastern Europe. The year 1616 was characterized by a long dry period that began in mid-April and lasted through the summer. The regional focus of this dry phase

was in the East but eventually spread to the rest of Central Europe. Overall, 1616 was remarkably dry from April until November, characterized by a very hot and extremely dry summer (Glaser, 2013). The extreme dryness of this particular year, throughout whole Europe, is well captured by our VPD reconstruction.

The year 1737 is part of a relatively wet period which is represented by the PDSI (Cook et al., 2015; Ionita et al., 2021) for Europe on average. The more humid conditions also fit with a comparatively low VPD in Central Europe as well as in the

Mediterranean region as shown in Figure 9. However, relatively high VPD anomalies are shown over northern Scandinavia, which are also represented on average by the PDSI (Cook et al., 2015; Ionita et al., 2021). For example, August 1737 was too cold and too wet in Central Europe and almost continuous rainy weather was reported in Germany in the second decade of August (Glaser, 2013).

The year 1741 is often associated with the Irish famine, which according to documentary and early instrumental data was

associated with low rainfall in spring and summer (Pauling et al., 2006). In our VPD reconstruction (Fig. 9 upper right), parts of the UK and Germany are also represented by a high VPD anomaly, while the highest VPD anomalies are shown for France and northern parts of Spain. That agrees with the PDSI (Cook et al., 2015), precipitation (Pauling et al., 2006), and temperature reconstruction (Luterbacher et al., 2004). In contrast, low VPD values over Scandinavia disagree with wet conditions in the PDSI (Cook et al., 2015). For example, in 1741 the northeast part of Germany experienced longer-lasting cooler phases, which



were also very dry in some regions. In July, this pattern essentially remained intact. Overall, there was a prolonged period of rain in central Germany, that lasted until the winter. In August numerous downpours and thunderstorms have been reported (Glaser, 2013).

As already shown in the temporal perspective of VPD variability, the year 1814 shows the lowest VPD value for the Mediterranean region (Fig. 9 lower left). However, there are high VPD anomalies in Scandinavia. The pattern resembles a dipole

pattern which shows similarities to the summer European blocking pattern (Barnston and Livezey, 1987; Cassou et al., 2005) which is often associated with the Summer North Atlantic Oscillation (SNAO; Hurrell and Van Loon, 1997). According to Cassou et al. (2005), 17.8 % of the positive phase and 17.9 % of the negative phase of the summer European blocking pattern influence the total summer weather regimes in Europe. Furthermore, the summer European blocking pattern is a surrogate indicator for storm track activities (Folland et al., 2009; Lehmann and Coumou, 2015). The SNAO pattern is also evident in

the OWDA for 1814.

The last selected year with humid conditions in Europe is 1816, which is related to the Tambora eruption in 1815 that caused a cooling of 2-3 °C in Central Europe (Trigo et al., 2009). Several studies refer to the year 1816 as the "year without a summer" due to a sharp drop in temperature revealed in European reconstructions (e.g., Büntgen et al., 2006). 1816 was an unusually cold and wet year, especially in Central Europe (Schurer et al., 2019). The largest temperature anomalies occurred in summer

1816, marking the coldest European mean summer temperature ever recorded over the last 235 years (Casty et al 2007). It was also associated with a Europe-wide famine and a sharp increase in food prices across Europe (Brönnimann and Krämer, 2016). In our study, Central and Western Europe show low VPD anomalies, while Southeast Europe and Scandinavia are interestingly not affected by a low VPD anomaly (Fig. 9). The described pattern agrees well with the pattern resulting from reconstructed temperatures, precipitation, and sea level pressure (e.g., Schurer et al., 2019).

Based on PDSI data (Cook et al., 2015), the conditions for 1921 are the driest of the last millennium, particularly for western and Central Europe. This drought was described as "a year of unprecedented low rainfall" across much of the British Isles, with the greatest deficit occurring in southeast England (Brooks and Glasspoole, 1922). 1921 was one of the driest years on record, over large areas in Europe, with a focus on Central Europe (Ionita and Nagavciuc, 2021) which is in agreement with the spatial pattern of our VPD reconstruction. This extreme dryness was mainly driven by a long-lasting precipitation deficit

throughout the year (Ionita and Nagavciuc, 2021). In addition, 1921 is also considered the driest year in the Rhine and Weser catchments based on observed runoff data (Ionita et al., 2021). This description fits the depicted conditions in Central and Western Europe, as large parts of Great Britain, Germany and Poland show high VPD anomalies in Figure 9 (lower right). Furthermore, it is evident from the time series that the VPD shows the second-highest value within the last 400 years for Central Europe (Fig. 5b). The less dry conditions in Italy are consistent with other studies (e.g. Bonacina, 1923).



### 3.5 Historical and future European VPD in CMIP6

We compare the reconstructed VPD reconstruction with the ability of the CMIP6 models (Eyring et al., 2016) to simulation the VPD in the historical simulations and assess how the VPD variability is projected to change in the 21st century. In this respect, we use the CMIP6 ensemble mean of VPD (see table S1 for a description of the used models) over the period 1851 – 2100 (merging ensemble averages from historical and future experiments). For Northern Europe, the model ensemble we have chosen represents on average VPD in the historical run too weakly (Fig. 10 up). Nevertheless, a large part of the observed and reconstructed VPD lies within the 75th percentiles of the model ensemble. From 1980 onwards, the model ensemble shows a positive trend of VPD until 2014 (m= 0.0014 kPa; p-value < 0.0000001), which cannot be represented by the reconstruction or observations. In the future scenarios, the strongest increase is shown by the scenario SSP5-8.5, whereas SSP1-2.6 and SSP2-4.5 show only a slight increase and small differences of VPD in Northern Europe (Table 2). For the last two scenarios, VPD does not increase beyond the observed values.

There is a significant offset between the historical run and the reconstructed as well as the observed VPD can be seen for Central Europe (Fig. 10 mid). In contrast to Northern Europe, VPD in Central Europe is higher than the model ensemble. However, the observed VPD is close to the 25th percentile of the model ensemble. This is an interesting result which notably indicates that CMIP6 models are modestly able to simulate the observed VPD in those regions. From 1980 onwards, the model ensemble shows an increasing trend of VPD until 2014 (m= 0.0051 kPa; p-value < 0.000000001), which is also represented by the reconstruction and observations (m= 0.0028 kPa; p-value < 0.01). Furthermore, all three scenarios show a significantly stronger increase than for Northern Europe (Table 2). It is noticeable that the SSP5-8.5 scenario provides the strongest VPD increase, and the other two scenarios behave very similarly despite a stronger increase compared to the projected changes of VPD in Northern Europe. SSP2-4.5 shows only minimal increased values compared to SSP1.2-6.

The strongest difference between observed and the modelled historical runs can be found for the Mediterranean region (Fig. 10 down). Thus, the observed and model data differ by more than 0.5 kPa. Furthermore, the observed data also lie outside the interquartile range. However, the ensemble mean shows a trend from 1990 until 2014 (m= 0.006 kPa; p-value < 0.000000001) which is also represented in the reconstruction and observations (m= 0.004 kPa; p-value < 0.01). As for Central Europe, the strongest VPD increase can be determined for the scenario SSP5-8.5 (Table 2). On average, we show an increase in VPD of more than 1 kPa by the end of the 21st century. This is the largest increase of all regions considered. Furthermore, there is a clearer difference between the other two scenarios. In the scenario SSP1.2-6 there is only a comparatively weak increase which levels off again after a short time. A much larger increase can be observed for scenario SSP2.4-5, where the trend is increasing until the end of the century. This cannot be found in the other regions.

Thus, in order to clarify the question of how VPD will develop spatially in the future and what causes the behaviour in the scenarios, Figure 11 shows the change in VPD in the three scenarios for the European continent during summer. It is noticeable that especially the Mediterranean region is a hotspot for the increase of VPD as already shown in Fig. 7. However, for the two regions of North and Central Europe, a different pattern is observed. For Central Europe, only a small and localized increase

in VPD can be seen in SSP1.2-6 whereas Northern Europe shows even a smaller increase of VPD in this scenario. High VPD values from the subtropics expand in these regions. This is detectable especially in Central Europe in the SSP2.4-5 scenario, where a stronger increase of VPD is observed. Nevertheless, a stronger and pronounced increase of VPD in Northern Europe can only be detected in scenario SSP5-8.5.

450

**Figure 10: Five centuries of VPD variability for three different regions in Europe.** The boundaries of the different regions are following the definitions of Iturbide et al. (2020). In this graph, we join the reconstructed (VPD), the observed (20CR) as well as the modelled VPD for the historical conditions (Hist) as well as for three different scenarios. (SSP1.2-6, SSP2.4-5, SSP5.8-5) Since we use a model ensemble for the presentation of observed VPD, we show the average with a bold line and the area between the 25[th] and 75[th] percentile with the corresponding slightly transparent colour.

455





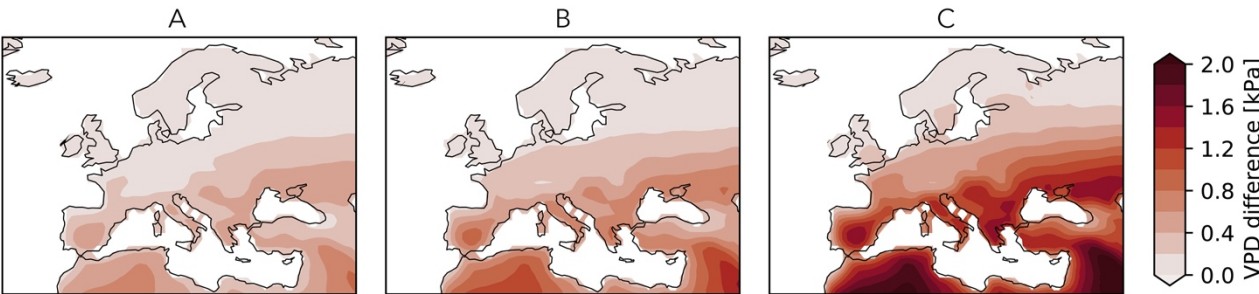

**Figure 11: Spatial distribution of the summer VPD anomalies in Europe based on the projected changes (2071-2100) for SSP1-2.6 (A), SSP2-4.5 (B), and SSP5-8.5 (C) scenarios relative to the historical summer VPD during the 1971–2000 reference period**. For the historical and projected changes, the average temperature of the CMIP6 model ensemble is used. All changes are significant according to the two-sided Student's t test ($p<0.05$).

**Table 2: Characteristics of the linear trend of three different future scenarios for Northern Europe, Central Europe, and Mediterranean.**

| | SSP1.2-6 | | | SSP2.4-5 | | | SSP5-8.5 | | |
|---|---|---|---|---|---|---|---|---|---|
| **NORTHERN EUROPE** | m= | 0.0003 | kPa | m= | 0.0005 | kPa | m= | 0.0014 | kPa |
| | p-value < 0.00001 | | | p-value < 0.000000001 | | | p-value < 0.000000001 | | |
| **CENTRAL EUROPE** | m= | 0.0008 | kPa | m= | 0.0022 | kPa | m= | 0.0083 | kPa |
| | p-value < 0.001 | | | p-value < 0.000000001 | | | p-value < 0.000000001 | | |
| **MEDITERRANEAN** | m= | 0.0013 | kPa | m= | 0.0051 | kPa | m= | 0.0134 | kPa |
| | p-value < 0.000001 | | | p-value < 0.000000001 | | | p-value < 0.000000001 | | |

## 4. Limitations of the reconstruction

The used $\delta^{18}O_{cel}$ network is characterized by specific limitations that influence the quality of the results presented in this study. An over-representation of the sample sites in Central and Western Europe compared to the ones from Southeast, East and Northern Europe is shown in Figure 1. These characteristics of the spatial distribution of sites are also represented in the validation statistics, where the regions with a good sample density show good validation scores (Section 3.2). Therefore, further time series from uncovered regions are needed to enhance the quality and the spatial extent of our reconstruction. More time series from covered regions wouldn't improve much the RF model, because they are expected to be strongly correlated with the already used time series, which would have a very slight influence on the predictive performance of the RF model (Boulesteix et al., 2012).

Beside the $\delta^{18}O_{cel}$ climate signal, the observational data of VPD is given by the ensemble mean of 20CRV3 (Slivinski et al., 2019) for the period 1900 to 1994. Even though the quality and quantity of instrumental data available during this period are comparatively good, the ensemble mean can only represent the variability and diversity of the reanalysis with 80 ensemble





members to a limited extent. Therefore, future studies are welcome to compare the quality of the reconstruction with other ensemble members.

In our study, the reconstruction is based on the application of the RF algorithm calibrated and evaluated over the period 1900 to 1994. The model is therefore trained to represent exactly this period. Thus, when this model is applied to the years prior, we assume stationarity between VPD variability and proxy records, as is generally the case for climate reconstructions (e.g., Cook et al., 2015). Since the climate system is not stationary, the assumption of stationarity must be included as a potential source of error. However, the RF approach can represent non-linearities which is not possible with the classical approaches, for ex-

ample, PCR (e.g., Cook et al., 2015). Therefore, in order to eliminate errors when using the reconstruction, we recommend a comparison with other available observational data, climate proxies, or reconstructions to quickly identify discrepancies.

Finally, even if the nested reconstruction approach is used by default for reconstructions (Luterbacher et al., 2004; Pauling et al., 2006; Cook et al., 2007, 2019; Freund et al., 2019) to cover the longest possible time range, it is important to check the quality of the model for the respective time range. Therefore, we recommend considering the validation scores for the consid-

ered time periods which are also uploaded in the repository.

**5 Conclusion & Outlook**

Here, we present the first gridded reconstruction of the European summer VPD over the past 400 years. We show projected VPD conditions in the European summer based on three different scenarios from the recent CMIP6 archive. The combination of the reconstructed, observed and projected variability allows conclusions about the past, present, and future VPD variability

changes in Europe. Moreover, this is the first study that uses a $\delta^{18}O$ network and modern machine learning algorithm to perform a spatial reconstruction of a climate variable.

The past variability of VPD is different for the three investigated regions: North Europe, Central Europe, and the Mediterranean. The lowest VPD values, as well as the lowest variability, are obtained for North Europe over the past 400 years. Also, in comparison to the other two regions, North Europe is showing the smallest increasing VPD trend in the reconstruction from

the mid of the 18th century. The highest VPD values for this region are reached in the mid of the 17th century. Central Europe and the Mediterranean region reveal stronger trends of increasing VPD (highest VPD on average and highest VPD variability) which can be explained by a precipitation decrease and a temperature increase.

Our results underline that the European VPD has increased over the last decades. Based on the obtained long-term perspective, we find that this increase in Europe has not started in 2000, but has already begun a few decades after the Late Maunder

Minimum with a simultaneous increase of the temperature in the mid 18th century.

The strong increase of the VPD at the end of the 20th and beginning of the 21st century is represented by the model ensemble used, although the magnitude of VPD increase partly differs. We suggest that the differences are based on the differences in regional temperature between observation and models. However, when looking at the projected trends for the VPD, it is noticeable that the modelled trends for SSP1.2-6 and SSP2.4-5 are close to the trends between 1980 and 2014 of the observed



VPD. In the historical context, however, these trends are unique in magnitude and persistence, as comparable increases in VPD do not exist and, if they do, do not last longer than 30 years. Our results imply that vegetation in Europe has been subject to an increase in VPD for a longer period of time, but that increase has been significantly amplified by recent climate change, especially in Central Europe and the Mediterranean region. The association between high VPD and a decline in tree growth (Eamus et al., 2013), higher forest mortality (Park Williams et al., 2013), higher incidence of droughts (Dai, 2013), a decline

in crop production (Zhao et al., 2017) and a higher incidence of forest fires (Seager et al., 2015) suggests that high VPD events will become more frequent as the average VPD continues to increase in the 21$^{st}$ century, with the magnitude being determined by the amount of greenhouse gas emissions.

The presented VPD reconstruction helps to visualize the local and regional impacts of the current climate change as well as to minimize statistical uncertainties of historical VPD variability. Furthermore, the interdisciplinary use of the data should be

emphasized, as VPD is a crucial parameter for many climatological processes. As a logical next step, the regional and temporal boundaries of the reconstruction can be extended by using more and longer $\delta^{18}$O time series from tree-ring cellulose. It is also possible to disentangle the influence of solar events on the VPD on the local, regional and continental scale.

**Contributions**

D.F.B conceived the ideas and designed the methodology together with S.M. on the basis of the R codes of S.M.. D.F.B

analysed the data, drafted and led the writing of the manuscript with significant inputs from S.M., M.I., V.N., G.H., M.F., G.H.S., D.N.S. and G.L. All authors contributed critically to the drafts and gave final approval for publication.

**Data and Code availability**

All data from CMIP6 simulations used in our analyses are freely available from the Earth System Grid Federation (https://esgf-data.dkrz.de/projects/cmip6-dkrz/). The ISONET network is not publicly available. The time series of the individual sample

sites can be request by the corresponding authors of the mentioned studies in the proxy network Section. The climate data from 20CRv3 is freely available. The postprocessing of the model and reanalysis output data has been done with the Climate Data Operators (Schulzweida, 2019) and the corresponding Python binding. Furthermore, the reconstruction of the VPD have been done with a modified version of the ClimIndRec version 1.0 scripts (https://zenodo.org/record/5716236#.YZkyyi2ZOL8) in R (version 4.0.2). The required packages (glmnet, pls, randomForest, ncdf4 and dependencies) are freely available. The gridded

VPD reconstruction, the VPD datasets for 20CR and the CMIP6 model ensemble can be downloaded from here: https://doi.org/10.5281/zenodo.5958836 (Balting, D. F. et al., 2022).



**Competing interests**

The authors declare that there is no conflict of interest.

**Acknowledgment**

Funding by the PalEX Project (AWI Strategy Fund) is greatly acknowledged. .GL and MI are supported by Helmholtz funding through the joint program "Changing Earth - Sustaining our Future" (PoF IV) program of the AWI. Funding by the Helmholtz Climate Initiative - REKLIM is gratefully acknowledged. VN  Viorica Nagavciuc was partially supported by a grant of the Ministry of Research, Innovation and Digitization, CNCS/CCCDI – UEFISCDI, project number PN-III-P1-1.1-PD-2019-0469, within PNCDI III.. All but four tree-ring stable isotope chronologies were established within the project ISONET sup-

ported by the European Union (EVK2-CT-2002-00147 'ISONET'). We want to thank all participants of the ISONET project (L. Andreu, Z. Bednarz, F. Berninger, T. Boettger, C. M. D'Alessandro, J. Esper, N. Etien, M. Filot, D. Frank, M. Grabner, M. T. Guillemin, E. Gutierrez, M. Haupt, E. Hilasvuori, H. Jungner, M. Kalela-Brundin, M. Krapiec, M. Leuenberger, H.H. Leuschner, N. J. Loader, V. Masson-Delmotte, A. Pazdur, S. Pawelczyk, M. Pierre, O. Planells, R. Pukiene, C. E. Reynolds-Henne, K. T. Rinne, A. Saracino, M. Saurer, E. Sonninen, M. Stievenard, V. R. Switsur, M. Szczepanek, E. Szychowska-

Krapiec, L. Todaro, K. Treydte, J. S. Waterhouse, and M. Weigl). The data from Turkey, Slovenia and Southwest Germany were produced with the EU-funded project MILLENNIUM (GOCE 017008-2'MILLENNIUM'), special thanks to T. Levanic and R. Touchan. The tree-ring stable isotope chronologies from Bulgaria were established with support of the German Research Foundation DFG (HE3089-1, GR 1432/11-1) and in cooperation with the administration of Pirin National Park, Bulgaria. Furthermore, we want to thank Paul Gierz and Christian Stepanek for technical support.

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
