# Peer review of "A past, present and future perspective on the European summer vapour pressure deficit"

_Earth System Science Data, 2022_

## Author Comment (AC1)

**Response reviewer #1**

Topic: VPD, as a climate index to measure drought, has attracted increasing attention. Moreover, it is of great scientific and guiding significance to reflect its spatio-temporal change pattern from a longer time scale.

Research content: The spatial distribution of VPD in European summer is analyzed in three-time scales. Based on the oxygen isotope records of the last 26 European tree rings, the VPD time series grid of the past 400 years was reconstructed based on the random forest. At the same time, the change of VPD at the end of 21st century is simulated based on CMIP6 data. Strong continuity of time is the main innovation of this article. Combining paleoclimate data and CMIP6 data, it expands the time range of existing research and has a strong guiding and demonstrating role. At present, the research on paleoclimate (Quaternary) mainly focuses on the reconstruction of paleoclimate. It is innovative to combine it with VPD. The simulation of VPD in the future is also based on the latest CMIP6 data, which has a certain scientific research opportunity. The idea of this paper is ingenious, and the content involves many research fields. Combining with the current hot VPD, this paper analyzes the spatial and temporal variation characteristics of VPD in Europe from the perspective of geography, which is frontier in the field and rich in content.

We would like to thank the reviewer for his/her appreciation and for taking the time to read and review our paper.

Questions: The paper is well done. However, the accuracy of VPD in paleoclimate reconstruction needs to be further improved. The paleoclimate temperature is retrieved from tree rings, and VPD is estimated based on paleoclimate. Moreover, there are only 26 station data, which may affect the inversion accuracy. In addition, there are no other related indicators to verify the paleoclimate of tree ring inversion. We can compare the paleoclimate based on pollen, foraminifera, stalagmites and other indicators on the same time scale to further verify the accuracy of tree ring inversion.

We thank anonymous reviewer #1 for his scientific evaluation of our manuscript. Moreover, we are pleased that reviewer #1 clearly emphasizes the scientific importance of our manuscript and acknowledges the methodological challenge of combining different topics. Indeed, we agree with #Reviewer #1 that the accuracy and validity of our reconstruction can be further improved. In principle, the limitations of the proxy network we use have been described and explored in detail in numerous other studies (e.g., Treydte et al., 2007b, a) and are evaluated in Section 4 in the context of our VPD reconstruction. In this respect, the climate signals of the used network have already been studied and compared with climate signals from other climate proxies over long timescales (e.g., Balting et al., 2021). However, a comparison with the climate proxies listed by reviewer #1 is difficult, as these have either not yet been associated with the complexity of the VPD variable or they have different temporal resolutions and a different seasonal/monthly signal which makes a statistical comparison difficult. Moreover, the study already bundles several complex research areas and further comparison with other proxies is beyond the scope of the study. In our study, the most important comparison remains between the isotopic network used and the climate observations, which we detail in Sections 3.1 and 3.2. Nevertheless, a more detailed comparison with other climate proxies is a very good idea for a potential fowling study.

**References**

Balting, D. F., Ionita, M., Wegmann, M., Helle, G., Schleser, G. H., Rimbu, N., Freund, M. B., Heinrich, I., Caldarescu, D., and Lohmann, G.: Large-scale climate signals of a European oxygen isotope network from tree rings, Clim. Past, 17, 1005–1023, https://doi.org/10.5194/cp-17-1005-2021, 2021.

Treydte, K., Schleser, G. H., Esper, J., Andreu, L., Bednarz, Z., and Berninger, F.: Climate signals in the European isotope network ISONET, TRACE, 5, 138–147, 2007a.

Treydte, K., Frank, D., Esper, J., Andreu, L., Bednarz, Z., Berninger, F., Boettger, T., D'Alessandro, C. M., Etien, N., Filot, M., Grabner, M., Guillemin, M. T., Gutierrez, E., Haupt, M., Helle, G., Hilasvuori, E., Jungner, H., Kalela-Brundin, M., Krapiec, M., Leuenberger, M., Loader, N. J., Masson-Delmotte, V., Pazdur, A., Pawelczyk, S., Pierre, M., Planells, O., Pukiene, R., Reynolds-Henne, C. E., Rinne, K. T., Saracino, A., Saurer, M., Sonninen, E., Stievenard, M., Switsur, V. R., Szczepanek, M., Szychowska-Krapiec, E., Todaro, L., Waterhouse, J. S., Weigl, M., and Schleser, G. H.: Signal strength and climate calibration of a European tree-ring isotope network, Geophys. Res. Lett., 34, 1–6, https://doi.org/10.1029/2007GL031106, 2007b.

---

## Author Comment (AC2)

**Response reviewer #2**

This manuscript is a concerted and cooperative effort for investigating the long-term trend in summer VPD over Europe throughout the past, present, and future. The Author utilised a network of tree-ring $\delta^{18}$ O chronologies for the first time to reconstruct the European summer VPD for the past four centuries. The Material and Methods have been clearly described and comprehensive analyses have been performed. In the results and conclusion, the author did an impressive job in investigating and discerning the spatial and temporal changes in VPD.  Concerning the hot topic and critical situation of the current changing climate, the content of this research is requisite and very practical for addressing the diverse effects of climate change on the ecosystem.

We thank anonymous reviewer #2 for his time and effort in reviewing our manuscript. Additionally, we would like to express our gratitude for the appreciative feedback.